# Diagnostic Findings and Surgical Management of Three Dogs Affected by Osseous Metaplasia Secondary to a Salivary Mucocele

**DOI:** 10.3390/ani13091550

**Published:** 2023-05-05

**Authors:** Matteo Olimpo, Erica Ilaria Ferraris, Lorenza Parisi, Paolo Buracco, Sara Gioele Rizzo, Davide Giacobino, Andrea Degiovanni, Lorella Maniscalco, Emanuela Morello

**Affiliations:** Department of Veterinary Sciences, University of Turin, 10095 Grugliasco, Italy; ericailaria.ferraris@unito.it (E.I.F.); lorenza.parisi@unito.it (L.P.); paolo.buracco@unito.it (P.B.); saragioele.rizzo@unito.it (S.G.R.); davide.giacobino@unito.it (D.G.); andrea.degiovanni@unito.it (A.D.); lorella.maniscalco@unito.it (L.M.); emanuela.morello@unito.it (E.M.)

**Keywords:** sialocele, mucocele, salivary glands, osseous metaplasia, dog, soft tissue surgery

## Abstract

**Simple Summary:**

A salivary mucocele is the most common disorder affecting the salivary glands. Saliva leakage from the salivary gland parenchyma and/or from associated duct damage causes a chronic inflammatory process which can occasionally result in osseous metaplasia. Dogs having an ossified sialocele present with a hard, non-fluctuating mass containing a viscous, sticky liquid at centesis. Either radiography or computed tomography can confirm the thick bone-like wall pseudocyst. Surgical excision of both the pseudocyst and the affected salivary gland represents the therapeutic gold standard in these cases. The aim of this study was to report the clinical and diagnostic findings, surgical management and histopathologic report of three cases of cervical sialocele complicated by pseudocapsule osseous metaplasia.

**Abstract:**

Saliva is an irritant of the subcutaneous tissue, thus causing the development of a non-epithelial reactive pseudocapsule. Metaplastic ossification of the pseudocapsule is a condition rarely described in the veterinary literature. The main causes of calcification are trauma, tumours, various chronic inflammatory conditions and *fibrodysplasia ossificans progressiva*. The aim of the present case series was to describe three dogs affected by a calcified salivary mucocele. The medical records of dogs affected by a cervical sialocele were retrospectively evaluated, and three cases met the inclusion criteria. All the dogs in this study were referred to the Veterinary Teaching Hospital (VTH) of the Department of Veterinary Sciences of the University of Turin (Turin, Italy) for a large solid mass in the intermandibular region. The diagnosis of a mucocele was confirmed clinically by centesis and by radiography or CT. Complete excision of both the pseudocyst and the ipsilateral mandibular/monostomatic sublingual salivary gland was performed in all cases. The histological report showed large areas of bone metaplasia within the pseudocapsule and chronic sialadenitis. Based on this limited case series, complete excision of the pseudocyst and a concurrent sialoadenectomy provided an effective treatment for this rare salivary mucocele disorder.

## 1. Introduction

Salivary gland (SG) disorders have rarely been described in dogs and cats, with only a 0.3% overall prevalence of veterinary consultations [1]. Salivary gland disorders in dogs include neoplasm, sialadenitis, salivary mucocele (commonly called sialocele) and various degenerative or fibrotic lesions [1]. A sialocele is an accumulation of saliva in the subcutaneous tissue with a consequent inflammatory reaction to saliva [2]. A sialocele is lined by inflammatory reactive connective tissue and not by epithelial tissue which, however, does occur in a pharyngeal cyst. Young male dogs appear to be those most represented; the breeds most affected are Poodles and German Shepherds; however, all breeds can be affected [3,4,5]. Trauma is considered to be the primary cause of a sialocele, thus explaining its prevalence in younger dogs. Other suggested causes are foreign bodies, sialoliths, haematomas, neoplasia and recent oral surgery. However, in the majority of cases, the real cause remains unknown [6]. A sialocele is classified as sublingual, cervical, pharyngeal or zygomatic. The sublingual/mandibular gland and duct complex is the most commonly affected structure [3]. The diagnosis is based on clinical presentation, history of onset and the result of the centesis. Tumours and abscesses may appear clinically similar, but they are generally either firm or painful. A thyroglossal cyst, a pharyngeal cyst and lymphadenopathy should also be considered when making a differential diagnosis [7,8,9,10].

The clinical presentation depends on the location: A cervical sialocele usually appears as a slowly enlarging nonpainful swelling in the intermandibular area. It can sometimes be difficult to differentiate the side of the affected gland involved in sialocele; if located on the midline, it may only slightly shift toward the originating side in a patient placed in dorsal recumbency. Centesis usually reveals a transparent viscous mucoid fluid, sometimes slightly bloody and/or cloudy. Periodic Acid–Schiff (PAS) staining can confirm this fluid as being saliva by revealing mucin [11,12].

A sialocele can be investigated using radiography/sialography, ultrasound, magnetic resonance imaging (MRI) and computed tomography (CT) [13,14,15,16]. Sialography has demonstrated the rupture of the duct/gland complex in 55% of cases [17]. Radiography usually reveals a uniform soft tissue swelling at the level of the cervical region while a fluid attenuating, non-contrast enhancing lesion with a smooth, soft tissue attenuating, contrast-enhancing wall is more often observed with CT. The size, shape and degree of loculation are variable [18].

Metaplastic ossification within the sialocele, for the most part at the level of its capsule, has previously been reported [19,20,21,22], and is likely due to chronic inflammation in the presence of granulation tissue.

The pseudocapsule of a salivary mucocele is microscopically visible as granulation tissue enclosing mucin; it is made up of edematous, vascularised connective tissue containing macrophages, lymphocytes and other chronic inflammatory cells.

In human medicine, an oral sialocele is a common SG disorder which appears more often on the lower lip mucosa or mandibular vestibule and is frequently caused by accidental lower lip biting. Bone metaplasia has also been reported in individuals with a caecal appendix [23,24] and a paranasal sinus mucocele [25,26].

In the past, the treatment of sialocele with drainage alone was characterized by a high rate of recurrence [13]. Surgical excision is the treatment of choice and involves the removal of the entire SG and ductus complex in order to avoid recurrence. At present, the need for complete pseudocapsule resection is under debate [2,11]. Sialoadenectomy can be performed using either a lateral or ventral [6] surgical approach; both approaches have been proven to be effective [27]. The sublingual/mandibular SG and duct complex can be removed bilaterally without risking a dry mouth as the production of saliva is ensured by other major and minor salivary glands [13].

No cases of human oral ossified sialocele have been reported to date. In veterinary medicine, only four case reports of sialocele-associated osseous metaplasia have been described [19,20,21,22].

## 2. Materials and Methods

### 2.1. Case Selection

The medical records of dogs referred to the Veterinary Teaching Hospital (VTH) of the Department of Veterinary Sciences of the University of Turin (Turin, Italy) for sialocele from January 2000 to January 2023 were identified and reviewed.

The inclusion criteria for this case series were as follows: (a) the presence of a mandibular/neck hard swelling; (b) a first or second level of diagnostic investigation to confirm the presence of a calcified lesion; (c) a sialoadenectomy, and complete or partial removal of the calcified lesion; (d) a complete histological report and (e) a minimum 30-day follow-up.

Written consent was obtained from the owners to proceed with anaesthesia and all the diagnostic and therapeutic procedures.

Perioperative standard-of-care management, including analgesia, was performed in all dogs. This study did not fall within the application areas of Italian Legislative Decree 26/2014, which governs the protection of animals used for scientific or educational purposes. Therefore, ethical approval was waived for this study. The animals were not treated as part of an experimental study; only the data were later selected and included in this study. No specific informed consent statement was obtained for inclusion in this retrospective study.

### 2.2. Diagnostic Imaging

The head and neck were investigated using CT (16 MDCT unit, Siemens Somatom Emotion) or radiographically (latero-lateral and ventro-dorsal views). For the CT scan evaluation, the dogs were premedicated with an intramuscular combination of dexmedetomidine (2 mcg/kg) (Dexdomitor^®^, Orion Pharma, Turku, Finland) and methadone (0.2 mg/kg) (Synthadon^®^, Le Vet Beheer B.V., Oudewater, The Netherlands). General anaesthesia was induced using propofol (3 mg/kg, IV) (Proposure^®^, Merial Italia Spa, Noventa Padovana, Italy) and was maintained with isoflurane in oxygen (Isoflo; Esteve Spa, Barcelona, Spain). The CT images were acquired in basal condition and after the intravenous administration of contrast medium Iomeron 400 mg/mL (1.5 mL/kg IV) (Bracco S.p.A., Milan, Italy).

### 2.3. Surgery

For patients undergoing a CT scan, the surgery was performed separately from the diagnostic imaging procedure in order to shorten the anaesthesia time.

After general anaesthesia, the dogs were clipped and moved to the operating theatre where they were positioned for the surgery and aseptically prepared. The intravenous administration of cefazolin (20 mg/kg) (Cefazolina Teva, Teva Italia SRL, Milano, Italy) was performed 20 min before skin incision and was repeated every 60 min until extubation. The affected SG was removed together with the entire ossified lesion. The suction of saliva was performed using a Poole suction tube. Both SGs and the pseudocapsule underwent histopathology; however, a portion of the pseudocapsule also underwent microbiological analysis. A passive drain was inserted at the surgeon’s discretion.

### 2.4. Histopathology

Both SGs and the pseudocapsule were fixed in 10% buffered formalin, decalcified for 48 h in Osteodec (Bio-Optica S.p.A., Milan, Italy), embedded in paraffin, sectioned at 4 μm and routinely stained with haematoxylin and eosin. A histological diagnosis was reached by the pathologists of the Department of Veterinary Science, University of Turin.

### 2.5. Postoperative Care and Follow-Up

Postoperative management included intravenous fluid (Lactated Ringer’s 2 mL/kg/h); analgesia with methadone (Semfortan^®^, Eurovet Animal Health B.V., Bladel, The Netherlands; 0.2 mg/kg IV q4h) or buprenorphine (Temgesic^®^, Schering-Plough, Segrate, Italy; 10 mcg/kg IV q8h); and meloxicam (Metacam^®^, Boehringer Ingelheim, Ingelheim/Rhein, Germany; 0.2 mg/kg IV the first day and 0.1 mg/kg IV thereafter). The dogs were discharged within 1–5 postoperative days with an Elizabethan collar and meloxicam (Metacam^®^, Boehringer Ingelheim, Ingelheim/Rhein, Germany; 0.1 mg/kg daily for seven days, PO), and were then re-evaluated weekly for a month after discharge. Complications were classified according to the guidelines of the “Veterinary Co-operative Oncology Group—Common Terminology Criteria for Adverse Events” (VCOG-CTCAE) [28]. The skin sutures were removed ten days post-operatively, and a first clinical evaluation was carried out. Additional follow-up information was acquired by periodic clinical evaluation or by direct communication with owners or the referring veterinarians.

## 3. Results

Three dogs presenting an ossified sialocele met the inclusion criteria and were included in this case series.

### 3.1. Case 1

A 3-year-old, 28 kg, intact female Boxer was referred to the VTH for a 6 cm painless firm swelling at the level of the right ventrolateral region of the neck. The swelling had been present for 3 months and had gradually grown. The dog showed no clinical signs of pain or discomfort except for drooling. Both the physical examination and laboratory profile (complete blood count, biochemical values and urinalysis) were within normal limits. There was no clinical evidence of regional lymphadenomegaly.

Sterile needle centesis was performed and revealed the presence of a stringy, brown fluid. Cytology of this fluid was consistent with saliva, owing to the presence of mucin aggregates, with no evidence of infection. Radiographic examination of the head and neck showed a large soft tissue consolidation with a calcified wall between the right intermandibular and subauricular regions. A skull CT scan was performed to identify the origin of this consolidation and its anatomical relationship with the surrounding structures. The examination revealed an ovoid capsulate mass (3.6 × 4.7 × 4.3 cm) at the level of the right pharyngeal region (Figure 1). The capsule was thickened (4 mm) and mineralised in its dorsal, lateral and medial aspects; it was thin and contrast enhanced in its ventral part. The content was fluid attenuating (around 25 HU) with no contrast enhancement. The structure was dorsal to the ipsilateral mandibular SG that appeared moderately compressed and dislocated cranially. Its medial aspect was in contact with the hyoid apparatus and nasopharynx, extending dorsally and nearing the tympanic bulla, while its lateral aspect was close to the angle of the mandible. The right medial retropharyngeal lymph node was slightly enlarged (2.5 × 1.3 cm), showing a heterogeneous contrast enhancement. The right lateral retropharyngeal lymph node (1.3 × 1 cm) was dislocated laterally and ventrally, close to the mandibular lymph nodes.

The dog was placed in left lateral recumbency, and a rolled towel was placed under the neck to elevate the surgical site. A lateral surgical approach to the right mandibular SG was performed. The cystic mineralised lesion was operated on using a 10 cm skin incision, from the ventral aspect of the vertical ear canal to the caudal portion of the ramus of the mandible, just over the swelling. The platysma and parotidoauricularis muscles were incised and reflected ventrally. The lateral portion of the calcified lesion was gently isolated from the surrounding tissue and completely excised (Figure 2).

Part of the capsule and its contents were sent for microbiological analysis. The right medial and lateral retropharyngeal lymph nodes were excised.

Subsequently, a mandibular/sublingual SG and duct sialoadenectomy was performed; the duct was ligated and transected as rostrally as possible at the level of the lingual nerve, which was spared. Using a combination of blunt and sharp dissection, the glandular tissue and its duct were dissected from the capsule and excised by tunnelling under the digastricus muscle. The surgical site was flushed with warm saline solution and a Penrose drain was placed within the subcutaneous space exiting ventral to the incision. The incision was closed routinely in three layers. Part of the capsule, the excised SG and lymph nodes were sent for histological evaluation.

The dog recovered uneventfully within a week. The bacteriological analysis of the pseudocapsule was negative for any bacterial growth.

Histologically, the SGs showed multifocal moderate lymphoplasmacytic chronic sialoadenitis with the interstitial spaces expanded by thick bundles of fibrocytes (fibrosis). The cystic lesion was lined by granulation tissue and small foci of an osseous metaplasia (Figure 3). The lymph nodes removed were characterised by hyperplastic lymphadenitis.

There was no sign of recurrence more than five years after surgery.

### 3.2. Case 2

A 1.5-year-old, 38 kg, intact male Italian Cane Corso was admitted for an 8 cm hard swelling just caudal to the left mandibular ramus, which had been present for 6 months and significantly increased in size during the previous 2 weeks. The consistency appeared solid at physical examination. The macroscopic aspect of the fluid obtained by centesis was compatible with saliva; however, cytology was inconclusive. The latero-lateral radiographic view of both the skull and the cervical region highlighted an irregularly rounded area of bone-like radiopacity at the level of the pharyngeal region superimposed to the hyoid apparatus (Figure 4). Both the physical examination and laboratory profile (complete blood count, biochemical values and urinalysis) were within normal limits.

Based on these findings, a suspicion of sialocele was formulated and a left mandibular/sublingual SG and duct excision was proposed. The patient was positioned in right lateral recumbency. A rolled towel was placed under the neck to elevate the surgical site. A vertical incision centred on the ossified lesion was performed caudally to the angular process of the mandible (Figure 5). The subcutis was undermined with blunt and sharp dissection, and the mandibular/sublingual SGs were identified between the linguofacial and maxillary veins. The capsule of the glands appeared to strictly adhere to the ventro-caudally located ossified lesion (Figure 5). The mandibular and the sublingual glands were dissected free from both the surrounding tissues and the sialocele. The salivary duct was followed rostrally and dissected; the lingual branch of the trigeminal nerve was identified and freed from the duct. The latter was then ligated and transected as rostrally as possible. Two ipsilateral mandibular lymph nodes were removed for histopathology as they were enlarged. The thick ossified capsule of the sialocele was then incised, revealing a brownish viscous content which underwent microbiological analysis together with a portion of the capsule. The sialocele was freed from the surrounding tissues and excised. The surgical site was flushed with warm saline solution and a Penrose drain exiting from a separate ventral incision was inserted. The incision was routinely closed in three layers. Part of the capsule, the excised SG and lymph nodes were submitted to histology. The drain was removed 3 days later. The dog healed without complications within 10 days and the sialocele never recurred and died 8 years later from causes not related to the sialocele. The bacteriological analysis was negative for any bacterial growth. Histological examination of the SGs showed a multifocal moderate lymphoplasmacytic chronic sialadenitis with moderate ductal hyperplasia and mild interstitial fibrosis. The cystic lesion surrounded an inner space containing amorphous eosinophilic material admixed with cellular debris (necrosis) and was composed of a fibrous capsule admixed with granulation tissue, foci of osseous metaplasia and aggregates and small mature lymphocytes. Histological examination of the mandibular lymph nodes revealed diffuse lymphoid follicular hyperplasia and foci of interstitial fibrosis.

### 3.3. Case 3

A 12-year-old, 8 kg, spayed female Dachshund was referred for a 5 cm painless swelling in the intermandibular region. The lesion had grown during the previous two weeks and appeared firmly attached to the surrounding tissues. The lesion had a bony/cartilaginous consistency; however, its ventral border appeared incomplete as several small openings were palpated. Due to the compression on the pharynx and the trachea, the dog presented with dyspnea on exertion. A small right sublingual sialocele was also present. Haematologic laboratory tests showed increased gamma-glutamyl transferase, bilirubin, total protein and hypoalbuminemia. These anomalies, associated with potbelly and bilateral and symmetrical back alopecia, were suggestive of hyperadrenocorticism, and an internal medicine specialist consultation was required. Low dose dexamethasone suppression test confirmed Cushing’s syndrome. Cytological analysis of the swelling was compatible with saliva; therefore, a calcified sialocele was clinically suspected.

A CT scan revealed a complex cystic formation extending from the right mandibular region, between the tongue and the mandible, up to the ipsilateral laryngeal region. The swelling had dimensions of 4.5 cm (width) × 5.8 cm (height) × 6.1 cm (length) with multiple concamerations and partially mineralised walls (approximately 2 mm thick); the content was fluid attenuating with no contrast enhancement (average attenuation of 20 HU). The medial aspect of the lesion was in contact with the hyoid apparatus and the larynx, which were dislocated and compressed to the left, while the caudal margin was in contact with the right mandibular lymph node and the facial vein. The swelling extended dorsally in the pharyngeal region, reaching the right tympanic bulla and the external carotid artery. The right mandibular SG appeared adherent to the lateral wall of the swelling, resulting in its appearing dislocated and compressed laterally. The right retropharyngeal lymph node was enlarged (2.2 × 1.9 cm) with dishomogeneous contrast enhancement and was in contact with the caudal aspect of the swelling (Figure 6).

The adrenal glands were enlarged bilaterally. The right measured 2.2 cm × 1.1 cm and the left 2.3 cm × 1 cm with dishomogenous contrast enhancement. No macroscopically visible pituitary gland abnormalities were found.

Routine diagnostic tests confirmed the suspicion of primary hyperadrenocorticism, and trilostane (Vetoryl^®^, Dechra Veterinary Production S.r.l., Northwich, UK) therapy at 1 mg/kg q24h was started. On the day of surgery, the dog was positioned in dorsal recumbency, with a towel placed under the neck to slightly elevate it. A skin incision in the intermandibular region centring on the ossified lesion was performed. The platysma muscle was incised, and the ventral aspect of the lesion was separated from the surrounding tissues (Figure 7). The caudolateral aspect of the sialocele was freed from the mandibular SG capsule. A branch of the lingual vein, which tightly adhered to the capsule of the lesion, was ligated and transected in order to proceed with the dissection. The thick capsule was incised, and the brownish fluid was aspirated (Figure 7). Part of the pseudo capsule and part of its contents were sent for bacteriological analysis. The ventral portion of the sialocele was excised using Mayo scissors, revealing a multiloculated cavity. Two stay sutures were medially and laterally positioned on the remaining wall of the capsule in order to facilitate its separation from the most dorsal tissues, which were tightly adhered (but not infiltrated) to the thyroid cartilage. With a combination of blunt and sharp dissection, the ossified sialocele was completely excised. The right mandibular/sublingual SGs and duct were then surgically excised (Figure 7).

The duct was tunnelled under the digastricus muscle and freed from the mylohyoideus muscle to get as close as possible to the sublingual caruncle. The enlarged right medial retropharyngeal lymph node was also excised. A Penrose drain exiting ventral to the surgical site from a small separate incision was inserted, and the wound was closed in three layers. The dog was discharged from the Intensive Care Unit of the VTH 3 days later after removal of the drain. Histological analyses were done on the pseudocapsule, the excised SG and lymph nodes.

At the clinical check-up one week after surgery, the sublingual sialocele appeared to have been completely resolved. The bacteriological analysis carried out on the pseudocapsule and aspirated fluid was negative for any bacterial growth. The dog underwent periodic check-ups for the management of Cushing’s syndrome. At the time of writing, twelve months after surgery, there was no recurrence of the sialocele.

The histological SG report showed a multifocal moderate granulomatous chronic sialadenitis. The pseudocapsule was constituted of fibroconnective tissue with abundant foci of osseous metaplasia while the medial retropharyngeal lymph node appeared to be hyperplastic.

## 4. Discussion

Sialocele is the most common SG disorder affecting the mandibular/sublingual gland or duct [29]. The diagnosis is relatively simple in the majority of cases as the aspiration of the non-painful swelling usually reveals a viscous, honey-coloured, clear or blood-tinged fluid consistent with saliva. Clinical suspicion is confirmed by cytology, which usually reveals a small to moderate number of non-degenerated nucleated cells and diffuse aggregates of mucin [2]. In the case series reported here, the clinical presentation was different from that usually found in sialoceles as palpation did not reveal a poorly defined and fluctuating swelling but a solid mass. On palpation, the calcified consistency was clearly perceptible at the level of the cranial cervical region or slightly more lateral. This condition widened the range of the differential diagnosis, thus including other head and neck lesions which could eventually calcify, such as tumours, inflammatory processes and congenital cysts. The differentiation between a congenital cyst and a sialocele can only be reached histologically by visualising a secretory epithelium in the inner layer of the capsule in the first case and the granulation tissue in the second one [10].

Three subtypes of ectopic ossification may occur in the body: osseous choristoma, defined as the presence of normal bone cells in an atypical location; heterotopic ossification, defined as mineralisations due to systemic disease and osseous metaplasia, in which osteoblasts originate from other mesenchymal cells, such as fibroblasts. It has been hypothesised that this change occurs in order to replace more stress-sensitive cells with more resistant ones in an adverse environment, such as that created by chronic inflammation [24,25,30,31] and trauma [32]. A sort of genetic predisposition for osseous metaplasia has also been hypothesized [19]. It is probable that the chronic inflammation caused by the saliva in contact with the soft tissues is the underlying cause of the sialocele calcification [21]. Calcifications have been widely reported in the veterinary literature and appear to be related to tumours, endocrinopathies and inflammatory lesions [33,34]. Tumours which can typically present calcification spots are pilomatricomas, benign follicular tumours located on the neck and trunk [33,35,36]. Calcifications have also been reported in prostatic adenocarcinoma, cortical adrenal gland tumours [37,38] and metastatic lymph nodes [39,40]. Calcification within a thyroidal mass is not very common [41,42], and is often regarded as malignancy. Some endocrinopathies such as hyperadrenocorticism may also predispose to calcium deposition, although mainly affecting the dermis [34]. However, for dog #3, affected by hyperadrenocorticism, it cannot be excluded that the ossification process of the sialocele was accelerated by her endocrine condition. *Calcinosis circumscripta* is a rare syndrome generating calcified lesions, typically at the level of the subcutis and tongue. The pathogenesis of this syndrome is poorly understood; however, the outcome is usually positive after surgical excision, when possible [43]. Although not common, calcification is also widely documented in extraparenchymal prostatic cysts [44,45,46]. Bigliardi et al. [36] reported the removal of a 25 cm solid extraparenchymal prostatic cyst having a 1 to 3 mm thick wall, composed of vascularised fibrous connective tissue with multifocal osseous metaplasia and dense connective tissue infiltrated by a mixed population of neutrophils and lymphocytes. As these histological features were similar to those reported for calcified sialoceles, it could be hypothesised that the mechanism of occurrence was the same and was related to chemical insult.

Any mass should first be sampled in an attempt to determine its nature [47]. In the cases presented herein, the calcification was not a limitation to the needle penetration as the metaplasia did not involve the entire pseudocapsule. Moreover, even in the case of complete calcification, a 2–3 mm capsule should be easily perforated by the needle. The macroscopic characteristics of the aspirated fluid were similar to those already described in previous studies regarding sialoceles [29,48,49].

In the literature, there are only sporadic case reports of dogs affected by ossified sialoceles [19,20,21,22]. Although the presence of ossification foci within the sialocele is a common occurrence in veterinary medicine [18], complete pseudocapsule ossification is rare. In two of the cases reported here, calcification foci were also found within the sialocele, adherent to its capsule.

Clinical presentation alone, similar to the unusual cases included in this case series, may not be enough to reach a diagnosis, identify the affected structures and plan the surgical procedure; in fact, diagnostic imaging is required. Radiography is able to confirm the calcification in the pseudocapsule but does not provide additional information, such as the origin of the lesion. Sialography is reported to have an accuracy of 66.7% in identifying the SG involved in the sialocele [50]; however, it has been almost completely replaced by more advanced imaging procedures which can also evaluate the extension of the lesion as well as its relationship with the surrounding tissues. A CT scan allows immediate and complete evaluation of the skull, neck and whole body when a complicated sialocele is suspected and a tumour and its metastasis have to be excluded.

Furthermore, when a sialocele is suspected, its location (right or left side, or bilateral) in relation to the position of the SGs should be evaluated. The suspected SG is generally contiguous to the fluid collection, with the affected gland which may appear atrophied, normal or enlarged at CT. The attenuation of SGs at CT may be normal or altered, sometimes with fatty infiltration or fluid pockets. These findings, associated with the presence of local fat stranding, may help in identifying the SG involved in the sialocele. Moreover, the need for advanced imaging in a preoperative setting is confirmed by the anatomical variations of the mandibular SG reported in the recent literature [51]. Although in some cases it is not possible to define which gland is leaking saliva, in the majority of cases, the collection occurs contiguously with the capsule of the affected gland. The latter may also appear as contrast enhanced with respect to the contralateral gland. A CT scan was available for two of the three cases reported herein, and the findings were suggestive of sialocele in both cases. In case #1, the SG origin clearly appeared to be the right sublingual/mandibular complex while in case #3, the lesion was at the level of the median sagittal plane, and both the sublingual/mandibular SGs appeared clinically and tomographically normal. For case #2, the marked lateralisation of the sialocele, together with its rostral extension, allowed clinically defining the side of the affected sublingual/mandibular SGs, even without the help of a CT scan which, at that time, was not available at the authors’ VTH. In all the cases reported herein, the sialocele originated from the sublingual/mandibular SGs and the corresponding ducts. This is in line with the literature which reports these structures to be those most frequently affected [3,52].

In a sialoadenectomy and duct excision regarding sublingual and mandibular SGs, both ventral and lateral approaches can be performed [2]. The ventral approach is preferred by some authors as it permits removal of the entire sublingual gland-duct complex [2]. It is associated with a lower risk of sialocele recurrence, and it is useful when a bilateral sialoadenectomy is planned. However, the ventral approach seems to be associated with an increased risk of wound-related complications [27]. Regarding the lateral approach, tunnelling under the digastricus muscle seems to increase salivary duct exposure, minimising the risk of leaving remnants of the rostral SG in situ [53]. In the cases reported herein, the choice of the surgical approach was, for the most part, based on the location of the saliva collection. In patient #3, a ventral approach was used to provide both a better visualisation of the sialocele and to excise as much of the salivary duct as possible, as a ranula was also present. On CT, both SGs appeared to be compressed by the lesion. The right sublingual/mandibular SG complex was excised as, after the positioning of the dog in dorsal recumbency, a right lateralisation of the sialocele was observed. For dogs #1 and #2, a lateral approach was used; however, the skin incision, instead of being horizontal and centred on the mandibular SG, was vertical and parallel to the vertical ramus of the mandible. The vertical incision allowed better visualisation of the pseudocapsule, which was separated from the surrounding tissues and was completely excised.

In the literature, a clear recommendation regarding the complete excision of the pseudocapsule is lacking [11,52]. In the cases presented herein, the sialocele was large, severely displacing important vascular and nervous structures as well as the trachea, causing breathing discomfort. For this reason and to avoid leaving calcified material in place, the pseudocapsule was completely removed in all cases, and drains were inserted to avoid seroma formation due to the extensive tissue dissection.

In all the dogs, the bacteriological analysis carried out on a sample of the pseudocapsule and its contents was negative, thus confirming the non-infectious origin of the lesion. Histological examination was performed in all cases, and it excluded a neoplastic process affecting the SG but, on the contrary, confirmed its inflammatory origin. Finally, the decision to perform an excisional biopsy of the tomographically or clinically enlarged lymph nodes can be a critical point; it was dictated by the fact that the unusual sialocele clinical presentation without the definitive histological diagnosis could not totally exclude a tumour.

The lymph node excision did not require additional skin incisions. The excised lymph nodes were adjacent to the glandular structures removed; therefore, their dissection did not lead to lengthening the surgical time or to an increase in post-surgical complications [11]. Hyperplastic reactive lymphadenitis was the definitive expected histological diagnosis in this case series, reflecting the tissue condition caused by both the space-occupying mass effect and the local inflammation.

The main limitations of this study were its retrospective nature and the small number of cases included, the latter reflecting the rarity of this condition. Another limitation was the different method of diagnostic imaging utilised as, for patient #2, a CT scan was not available.

## 5. Conclusions

The evidence of a huge and consolidated mass at the level of the neck should not immediately be considered highly malignant, and a complex sialocele should be considered in the differential diagnosis. Cytology, centesis and imaging are necessary to achieve the diagnosis and to plan the surgical approach. Ossified sialocele excision is associated with a favourable prognosis if the affected SGs are removed together with a part of or the entire pseudocapsule. Definitive diagnosis is provided by the histological examination.

## Figures and Tables

**Figure 1 animals-13-01550-f001:**
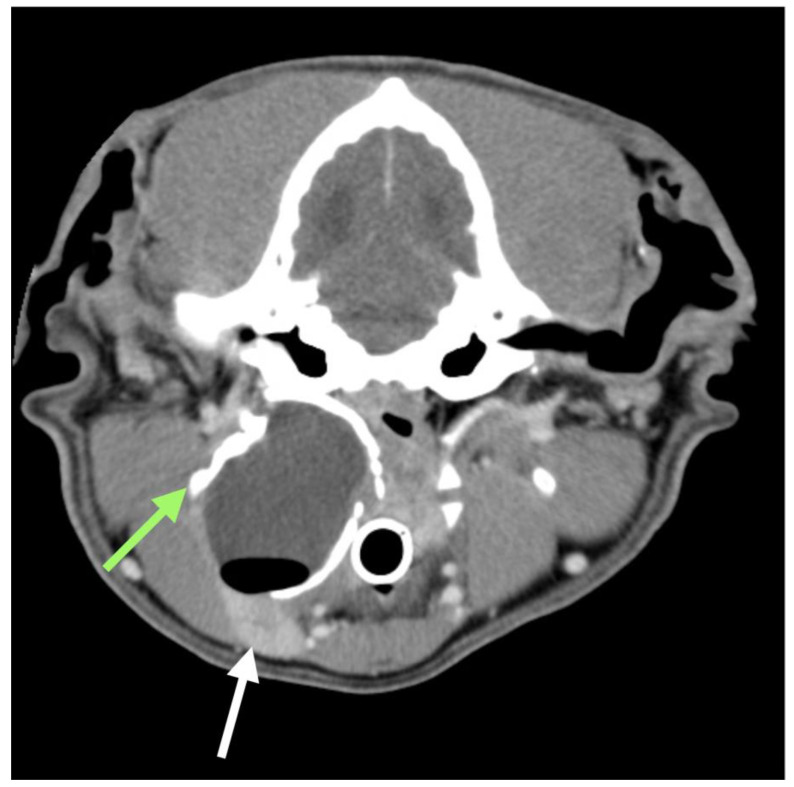
Traverse CT scan post contrast in case #1. The white arrow highlights the right mandibular salivary gland, while the green arrow indicates the mineralized wall in the dorsal aspect of sialocele. The presence of air in the cavity was secondary to fine needle aspiration.

**Figure 2 animals-13-01550-f002:**
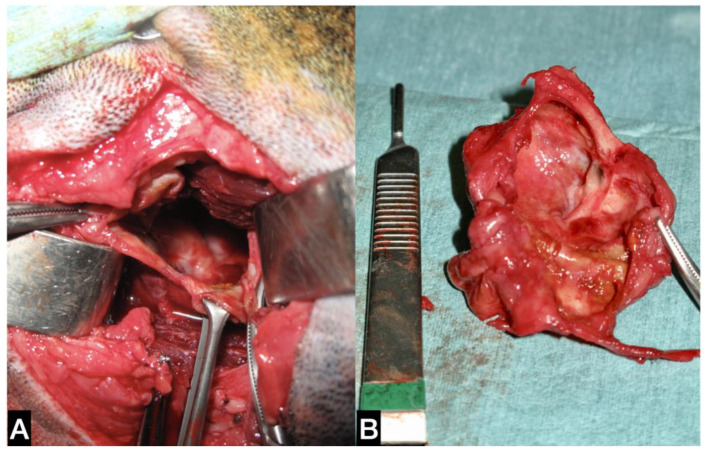
(**A**) Intra-operative picture of the lateral approach for mandibular/sublingual gland complex sialoadenectomy. The cystic cavity was opened in order to be emptied and then removed. Figure orientation: dorsal side—top figure; cranial side—right of the figure. (**B**) Macroscopic aspect of the opened cystic cavity.

**Figure 3 animals-13-01550-f003:**
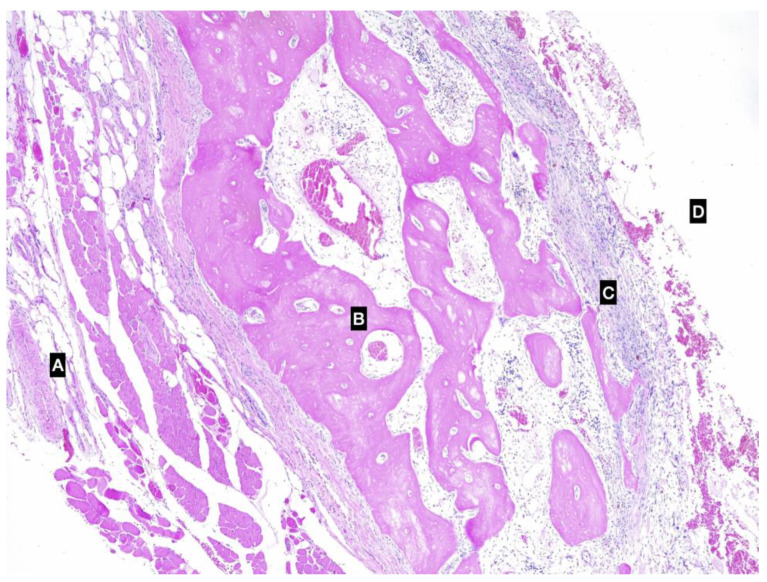
Photomicrographs of the ossified cystic cavity. (**A**) external aspect of the pseudocyst, muscular layer; (**B**) bone tissue; (**C**) connective tissue of the pseudocapsule; (**D**) inner cavity. Imaging was conducted using a haematoxylin and eosin stain.

**Figure 4 animals-13-01550-f004:**
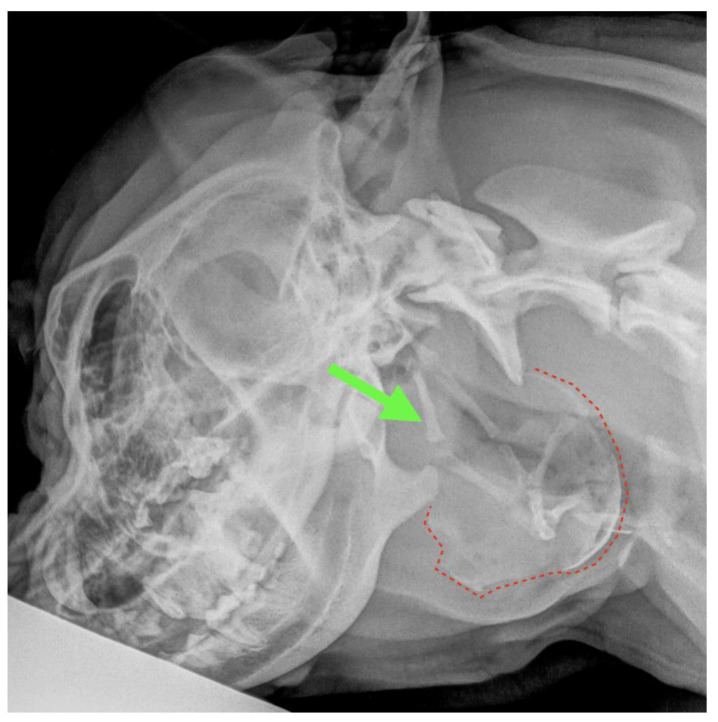
Latero-lateral radiographic projection of the head–neck. Green arrow: hyoid apparatus. Red dotted line highlights the presence of the ossified pseudocapsule of the sialocele.

**Figure 5 animals-13-01550-f005:**
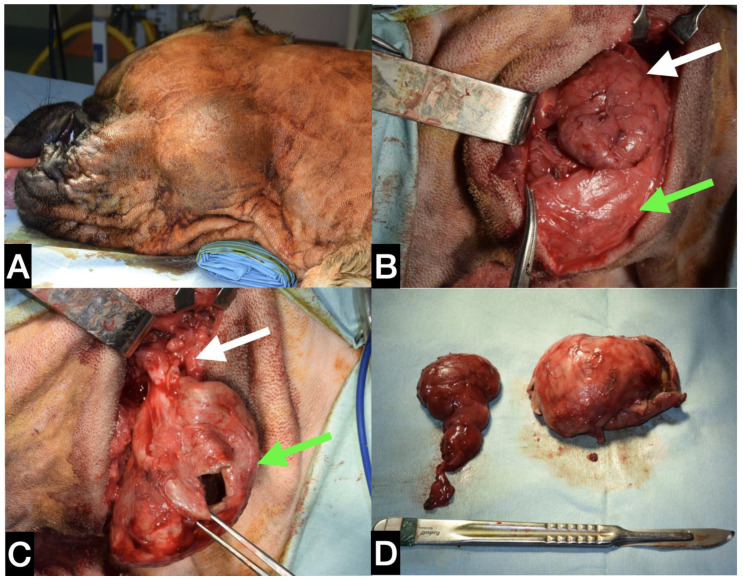
(**A**) Patient positioning in lateral recumbency; with a rolled towel positioned under the neck, the mandibular swelling is clearly evident. (**B**) Lateral approach for mandibular/sublingual complex sialoadenectomy. (**C**) A square fenestration was performed in order to remove the viscous fluid within the pseudocapsule. (**D**) Mandibular/sublingual complex and pseudocapsule. White arrow: mandibular salivary gland. Green arrow: ossified sialocele.

**Figure 6 animals-13-01550-f006:**
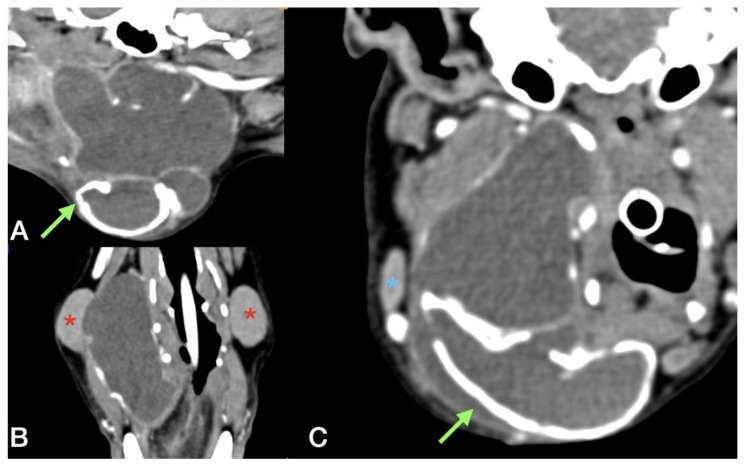
Multiplanar reconstruction of post contrast CT scan in dog #3. (**A**,**C**) Sagittal and transverse reconstruction: the green arrow underlines the thick and mineralized wall of the sialocele, while the blue asterisk indicates the right mandibular lymph node. (**B**) Dorsal reconstruction: the red asterisk indicates the mandibular salivary gland. The right salivary gland appears in continuity with the sialocele.

**Figure 7 animals-13-01550-f007:**
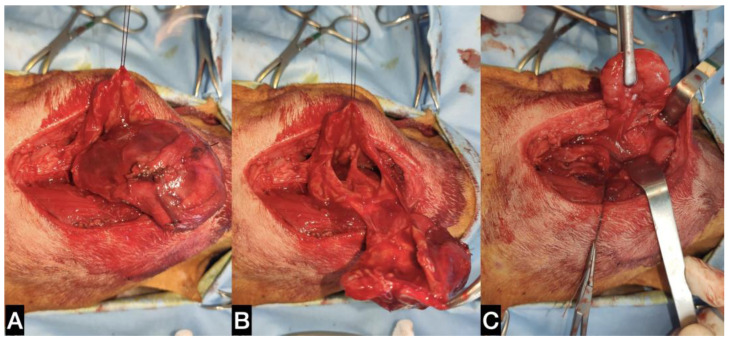
(**A**) Ventral approach to the sialocele. (**B**) The cystic cavity was opened and a stay suture was applied in order to retract the capsule. (**C**) After sialocele removal, a standard ventral sialoadenectomy has been performed.

## Data Availability

The authors declare that the data found in this paper are accessible and available.

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
