# Peer review of "Diagnostic Findings and Surgical Management of Three Dogs Affected by Osseous Metaplasia Secondary to a Salivary Mucocele"

_animals, 2023, doi:10.3390/ani13091550_

Round 1
Reviewer 1 Report
Dear authors thank you fo submitting your case report to this journal.
This short communication adds some interesting aspects on bone metaplasia, a rare occurrence in sialocele
Below is my review
The Simple Summary and Abstract are well-written and provide sufficient and clear information regarding the subject matter
Introduction:
lines 42-43 A sialocele is an accumulation of saliva in the sub-cutaneous tissue with a consequent inflammatory reaction to saliva [2,3].
Biblio 3 is no pertinent please delete
lines 44-46 Young male dogs appear to be those most repre sented; the breeds most affected are Poodles and German Shepherds; however, all breeds can be affected [4].
the bibliography cited is 30 years old, probably there are other studies in the literature that are more recent, I would ask you to include them
lines 53-56 Tumours and abscesses may appear clinically similar but they are generally either firm or painful. A thyroglossal cyst, a branchial cyst, and megalic lymph nodes have also should be be considered when making a differential diagnosis.
please insert biblio
lines 57-63 The clinical presentation depends on the location: a cervical sialocele usually appears as a slowly enlarging nonpainful swelling in the intermandibular area. It can sometimes be difficult to differentiate the side of the affected gland involved in sialocelewhich side of the affected gland involves the cervical sialocele; if located on the midline, it may only slightly shift toward the originating side in a patient placed in dorsal recumbency. Centesis usually reveals a transparent viscous mucoid fluid, sometimes slightly bloody and/or cloudy. Periodic Acid-Schiff (PAS) staining can confirm this fluid as being saliva by revealing mucin.
please insert biblio
Managing a sialocele using only drainage is characterised by a high recurrence rate [6].
This statement refers to treatments that are no longer in use. In my opinion it would be preferable to write for example: in the past, the treatment of sialocele with drainage alone was characterized by a high rate of recurrence...
Materials and Methods
lines 144-146 analgesia with methadone (Semfortan®, Eurovet Animal Health B.V., Bladel, The Netherlands; 0.5 mg/kg IV q8h) or buprenorphine (Temgesic®, Schering-Plough, Segrate, Italy; 0.3 mcg/kg IV q12 h). The analgesia protocol does not seem right to me.
The dosage of methadone, a pure agonist, might be OK, but the timing is not right. The animal remains uncovered for about 2-3 hours. As for buprenorphine, the dosage is not correct for a partial agonist/antagonist and the timing should not exceed 8 hours. Please explain or correct
Discussion
lines 362-363 Sialocele is the most common SG disorder affecting the mandibular/ sublingual gland or duct [22]
please insert newer bibliography
Conclusion
ok!
Author Response
Dear Editors and referees. The authors wish to collectively thank you for all of your constructive criticism of our manuscript. We appreciate the time and effort each of you have spent on this. We have made every attempt to address each point and will detail our responses and changes, when made, below. We believe the end result is certainly a stronger body of work thanks to your invaluable input.
Referee 1:
- lines 42-43 A sialocele is an accumulation of saliva in the sub-cutaneous tissue with a consequent inflammatory reaction to saliva [2,3].
Biblio 3 is no pertinent please delete
The reference has been removed
- lines 44-46 Young male dogs appear to be those most represented; the breeds most affected are Poodles and German Shepherds; however, all breeds can be affected [4].
the bibliography cited is 30 years old, probably there are other studies in the literature that are more recent, I would ask you to include them.
We added: Ortillès 2020 (DOI: https://doi.org/10.2460/javma.257.8.826 ); Lieske 2022(https://doi.org/10.1177/1040638720932169)
- lines 53-56 Tumours and abscesses may appear clinically similar but they are generally either firm or painful. A thyroglossal cyst, a branchial cyst, and megalic lymph nodes have also should be be considered when making a differential diagnosis.
please insert biblio.
We added: Bush 2023 (https://doi.org/10.1111/vsu.13928); Skinner 2021(https://doi.org/10.1111/vsu.13549); Markou 2020 (https://doi.org/10.1136/vetreccr-2019-000996), Nelson 2012 (https://doi.org/10.5326/JAAHA-MS-5728). We changed “branchial” to “pharyngeal” accordingly to Nelson 2012.
- lines 57-63 The clinical presentation depends on the location: a cervical sialocele usually appears as a slowly enlarging nonpainful swelling in the intermandibular area. It can sometimes be difficult to differentiate the side of the affected gland involved in sialocele which side of the affected gland involves the cervical sialocele; if located on the midline, it may only slightly shift toward the originating side in a patient placed in dorsal recumbency. Centesis usually reveals a transparent viscous mucoid fluid, sometimes slightly bloody and/or cloudy. Periodic Acid-Schiff (PAS) staining can confirm this fluid as being saliva by revealing mucin.
please insert biblio
We added: Ritter 2006 (https://doi.org/10.1080/00480169.2006.36720); MacPhail 2023, MacPhail, C. M. (2023). Disorders of the salivary gland. Small animal soft tissue surgery, 1-7.
- Managing a sialocele using only drainage is characterised by a high recurrence rate [6].
This statement refers to treatments that are no longer in use. In my opinion it would be preferable to write for example: in the past, the treatment of sialocele with drainage alone was characterized by a high rate of recurrence...
We changed the sentence as suggested.
- Materials and Methods
- lines 144-146 analgesia with methadone (Semfortan®, Eurovet Animal Health B.V., Bladel, The Netherlands; 0.5 mg/kg IV q8h) or buprenorphine (Temgesic®, Schering-Plough, Segrate, Italy; 0.3 mcg/kg IV q12 h). The analgesia protocol does not seem right to me.
The dosage of methadone, a pure agonist, might be OK, but the timing is not right. The animal remains uncovered for about 2-3 hours. As for buprenorphine, the dosage is not correct for a partial agonist/antagonist and the timing should not exceed 8 hours. Please explain or correct.
We changed the methadone and buprenorphine administration timing and dosage that were incorrect.
- Discussion
- lines 362-363 Sialocele is the most common SG disorder affecting the mandibular/ sublingual gland or duct [22]
please insert newer bibliography
We added: MacPhail 2023; MacPhail, C. M. (2023). Disorders of the salivary gland. Small animal soft tissue surgery, 1-7.
Reviewer 2 Report
It´s an interesting case series well written and discussed that merits its publication in order to divulge scientists and clinical veterinarians.
Some minor mistakes should be corrected:
line 55 have also should be be considered CORRECT
should also be considered
line 59 sialocelewhich INCLUDE SPACE
line 58 It can sometimes be difficult to differentiate the side of the affected gland involved in sialocelewhich of the affected gland involves the cervical sialocele REPHRASE
it can sometimes be difficult to differentiate the side of the affected gland involved in sialocele
line 55 megalic PERHAPS WOULD BE BETTER CHANGE THIS ADJECTIVE
line 194 A lateral surgical approach to the right mandibular SG
REPHRASE. This phrase doesn't have a verb
line 392 malignanciy MALIGNANCY

It´s an interesting case series well written and discussed that merits its publication in order to divulge scientists and clinical veterinarians.
The clinical cases are fully reported with good images. The matter is well presented in the introduction with appropriate scientific literature. After a full and well-constructed discussion the conclusions report an interesting and rare diagnosis and its apropriate management.
Author Response
Dear Editors and referees. The authors wish to collectively thank you for all of your constructive criticism of our manuscript. We appreciate the time and effort each of you have spent on this. We have made every attempt to address each point and will detail our responses and changes, when made, below. We believe the end result is certainly a stronger body of work thanks to your invaluable input.
Referee 2
- line 55 have also should be be considered CORRECT
should also be considered
Corrected.
- line 59 sialocelewhich INCLUDE SPACE
Space included.
- line 58 It can sometimes be difficult to differentiate the side of the affected gland involved in sialocelewhich of the affected gland involves the cervical sialocele REPHRASE
it can sometimes be difficult to differentiate the side of the affected gland involved in sialocele
Rephrased as suggested.
- line 55 megalic PERHAPS WOULD BE BETTER CHANGE THIS ADJECTIVE
We changed megalic to “lymphadenopathy”
- line 194 A lateral surgical approach to the right mandibular SG
REPHRASE. This phrase doesn't have a verb
We added “was performed”
- line 392 malignanciy MALIGNANCY
We correct the noun.
Reviewer 3 Report
Dear Authors,
this is a really interesting case series on a not so common pathology that deserves scientific investigation.
the structure of the article is clear, precise and exhaustive; the description of the clinical cases provides the necessary data aimed at understanding the pathology described, with a good quality of images.
I have only a few small questions that I write below:
Introduction
Line 59: please, remove “the side of the affected gland involved in sialocele”, this is probably a repetition of the sentence.
Discussion
Line 383: please, modify the format of the references in brackets “[24-27]”.
Figure 5. Caption: “(C) A square fenestration was performed in order to remove the complete pseudo-capsule.” You mean, the fenestration was performed to remove the viscous sialocele? It is not clear what you stated in the caption, because in the surgical description of the clinical case the whole capsule was removed. Please clarify.
Author Response
Dear Editors and referees. The authors wish to collectively thank you for all of your constructive criticism of our manuscript. We appreciate the time and effort each of you have spent on this. We have made every attempt to address each point and will detail our responses and changes, when made, below. We believe the end result is certainly a stronger body of work thanks to your invaluable input.
Introduction
Line 59: please, remove “the side of the affected gland involved in sialocele”, this is probably a repetition of the sentence.
We rephrased as follow: “it can sometimes be difficult to differentiate the side of the affected gland involved in sialocele”
Discussion
Line 383: please, modify the format of the references in brackets “[24-27]”.
The referenced papers are from 24 to 27, in this case the format is appropriate
Figure 5. Caption: “(C) A square fenestration was performed in order to remove the complete pseudo-capsule.” You mean, the fenestration was performed to remove the viscous sialocele? It is not clear what you stated in the caption, because in the surgical description of the clinical case the whole capsule was removed. Please clarify.
Yes, the fenestration was performed in order to remove the fluid, we corrected the sentence.